# Design, Characterization, and Biological Activities of Erythromycin-Loaded Nanodroplets to Counteract Infected Chronic Wounds Due to *Streptococcus pyogenes*

**DOI:** 10.3390/ijms24031865

**Published:** 2023-01-18

**Authors:** Narcisa Mandras, Anna Luganini, Monica Argenziano, Janira Roana, Giuliana Giribaldi, Vivian Tullio, Lorenza Cavallo, Mauro Prato, Roberta Cavalli, Anna Maria Cuffini, Valeria Allizond, Giuliana Banche

**Affiliations:** 1Department of Public Health and Pediatric Sciences, University of Torino, 10126 Turin, Italy; 2Department of Life Sciences and Systems Biology, University of Torino, 10126 Turin, Italy; 3Department of Drug Science and Technology, University of Torino, 10126 Turin, Italy; 4Department of Oncology, University of Torino, 10124 Turin, Italy

**Keywords:** *Streptococcus pyogenes*, chitosan, nanodroplets, erythromycin, skin and wound infections

## Abstract

*Streptococcus pyogenes* causes a wide spectrum of diseases varying from mild to life threatening, despite antibiotic treatment. Nanoparticle application could facilitate the foreign pathogen fight by increasing the antimicrobial effectiveness and reducing their adverse effects. Here, we designed and produced erythromycin-loaded chitosan nanodroplets (Ery-NDs), both oxygen-free and oxygen-loaded. All ND formulations were characterized for physico-chemical parameters, drug release kinetics, and tested for biocompatibility with human keratinocytes and for their antibacterial properties or interactions with *S. pyogenes.* All tested NDs possessed spherical shape, small average diameter, and positive Z potential. A prolonged Ery release kinetic from Ery-NDs was demonstrated, as well as a favorable biocompatibility on human keratinocytes. Confocal microscopy images showed ND uptake and internalization by *S. pyogenes* starting from 3 h of incubation up to 24 h. According to cell counts, NDs displayed long-term antimicrobial efficacy against streptococci significantly counteracting their proliferation up to 24 h, thanks to the known chitosan antimicrobial properties. Intriguingly, Ery-NDs were generally more effective (10^4^–10^3^ log_10_ CFU/mL), than free-erythromycin (10^5^ log_10_ CFU/mL), in the direct killing of streptococci, probably due to Ery-NDs adsorption by bacteria and prolonged release kinetics of erythromycin inside *S. pyogenes* cells. Based on these findings, NDs and proper Ery-NDs appear to be the most promising and skin-friendly approaches for the topical treatment of streptococcal skin infections allowing wound healing during hypoxia.

## 1. Introduction

*Streptococcus pyogenes* belongs to the serological group A of the streptococci (group A Streptococcus, GAS), is a clinically relevant bacterium and is one of the most common strictly human pathogens [1,2,3]. *S. pyogenes* results in a relevant amount of diseases globally and determines an important burden to national healthcare systems worldwide [4,5,6]. A large data set highlighted the key status of *S. pyogenes* between bacterial pathogens and the remarkable documentation of its impact on worldwide mortality and morbidity. Its infection rate is well documented; ~18.1 million people suffered from GAS diseases and these diseases are responsible for over 500,000 deaths each year [4,7,8,9]. *S. pyogenes* causes a wide spectrum of diseases varying from mild (i.e., impetigo, scarlet fever, tonsillopharyngitis) to life threatening (i.e., cellulitis, pneumonia, puerperal sepsis, streptococcal toxic shock syndrome, endocarditis, and necrotizing fasciitis) [2,3]. In addition, GAS infection can trigger serious post-infectious immune-mediated disorders and it is asymptomatically carried in the throats of healthy persons [3,10].

*S. pyogenes* cell surface displays several proteins and lipoteichoic acid that are important for the colonization of host tissues, the latter provides initial adherence of these bacteria to host surfaces by weak hydrophobic interactions, thereafter, other outer structures—such as, pili and lectin-carbohydrate interaction–further strengthen the binding [2,11,12]. Additionally, it produces several factors that may contribute and exacerbate the pathogenesis, that are extracellular enzymes such as proteinases, streptokinase, hyaluronidase, and neuraminidase, and toxins such as streptolysins, pyrogenic exotoxins, and streptococcal superantigens, some of which induce fever and shock [9,13,14]. After initial attachment to the host surface, *S. pyogenes* have been observed to form microcolonies; when bacterial cells proliferate, these microcolonies form complex structures, referred as a biofilm. It plays an important role in GAS pathogenesis [2]. In fact, these macroscopic structures have been involved in acute bacterial tonsillo-pharyngitis and streptococcal virulent wound and soft tissue infections. The emergence of antibiotic resistance among *S. pyogenes* has become a major concern globally, and one of its virulence properties—biofilm—has made it more resistant to antibiotic therapy and immune responses [15,16,17]. 

Among the therapeutic approach to fight *S. pyogenes* in skin, wound, and upper respiratory tract infection, penicillin remains the milestone, thanks to its efficacy, good safety profile, and low cost. Up to now, no confirmed *S. pyogenes* isolates were resistant to β-lactams. For subjects allergic to penicillin, erythromycin and other macrolides are prescribed as alternative treatments [18].

Nowadays, to treat infected chronic wounds, medical practitioners can find help in the nanotechnologies. Thanks to this it is possible to obtain controlled drug release, to minimize drug-associated adverse events, and to improve the drug performance [19,20,21,22].

Nanodroplets, constituted of chitosan as biocompatible and biodegradable polysaccharide known for its antimicrobial activity, are thought to be among the best candidate to encounter these characteristics as previously partially demonstrated [23,24,25]. The mechanisms of action of chitosan nanoparticles towards microorganisms are not still fully elucidated, however several authors reported that they can be adsorbed by the bacteria and, once internalized, they affect different key pathways for bacterial survival (i.e., disruption of cytoplasmic membrane and alteration of permeability, leaking of the cytoplasmic constituents, impair of mRNA synthesis and block of proteins activity, altogether resulting in cell death) [26,27,28,29,30,31].

On these grounds, the aim of this work was the design and characterization of low weight chitosan-nanodroplets (LW-cNDs, from now NDs), oxygen free (OF), or oxygen loaded (OL) containing erythromycin to be applied, as medical devices, for promoting healing of infected skin wounds. To achieve these goals, the prepared NDs were further assayed for their erythromycin release kinetics and biocompatibility for human keratinocytes. Thereafter, they were investigated for the potential uptake on *S. pyogenes* cells along with their inhibition rate on bacterial growth.

## 2. Results

Stable nanodroplet formulations, either oxygen free or oxygen loaded, for the delivery of erythromycin were obtained. The transmission electron microscopy (TEM) analysis of all NDs showed that they were spherical in shape with smooth surfaces, and no differences were noted when erythromycin was loaded into the OF or OLNDs (Figure 1). In Table 1, the physico-chemical properties of all the ND formulations are detailed. NDs sized in the nanometer range, with average diameters of 420 nm. Furthermore, they displayed positive zeta potentials with values around +31 mV and polydispersity index of ~0.23 (Table 1). Erythromycin (1.375 µg/mL) was efficiently loaded in NDs, either OF or OL, with a good encapsulation efficiency (98 and 96%, respectively). The addition of erythromycin did not interfere with their physico-chemical features, as reported in Table 1.

The in vitro drug release profile from the erythromycin loaded OFNDs and OLNDs is reported in Figure 2. No initial burst effect was revealed (Figure 2a), confirming the drug incorporation in the ND inner core, whereas a prolonged release kinetics of erythromycin from the NDs was demonstrated for all of the drug loaded formulations (Figure 2b).

Since the NDs were designed for skin wound healing purposes, the evaluation of the potential influence of both OFNDs and OLNDs loaded or not with erythromycin on the viability of eukaryotic human cells, specifically keratinocytes, was performed by the lactate dehydrogenase (LDH) tests. As depicted in Figure 3, neither OFNDs nor OLNDs displayed cytotoxicity on human keratinocytes (HaCaT cells), even when the test was achieved in normoxia or in hypoxia. It is worth noting that the addition of erythromycin did not affect eukaryotic cell survival since, also in this case, Ery-OLNDs and Ery-OFNDs were non-toxic for human keratinocytes. The percentages of cytotoxicity, in all the experiments, were less than 10% (Figure 3).

The microbiological experiments were performed on *S. pyogenes*, as representative bacteria causing wound infections by confocal microscopy and by the determination of the bacterial proliferation in presence of all the ND formulations [15,16,17].

As emerged from the analysis by confocal microscopy, OFNDs and OLNDs appeared to be already uptaken and internalized by bacteria at the earliest observational time-point (3 h of incubation) (Figure 4). The ND internalization by *S. pyogenes* was also observed at the latest time-point (24 h) of incubation, intriguingly, the loading of erythromycin into the NDs did not interfere with their internalization by streptococci. 

Finally, to assess the anti-streptococcal activity of both OFNDs and OLNDs loaded or not with erythromycin, *S. pyogenes* was incubated for different incubation times (4 h, 6 h, and 24 h) with the different ND formulations and then, at each timepoint, the cell count was determined (as Log_10_ CFUs/mL ± SEM).

As detailed in Figure 5, both OLNDs and OFNDs manifested long term antimicrobial efficacy against *S. pyogenes* since they significantly (*p* < 0.0001) counteract bacterial growth up to 24 h, displaying a slight bactericidal behavior due to the well-known chitosan antimicrobial properties [24,25,27,28,32,33,34]. After 4 h and 6 h of incubation, both OLNDs and OFNDs loaded with Ery significantly (*p* < 0.001) reduced *S. pyogenes* proliferation, respect to controls and to free-Ery, thereafter, at 24 h of incubation, both Ery-NDs and free-Ery revealed a total killing activity against the bacterium (*p* < 0.0001) (no bacterial colony was observed). Of note, at the earliest timepoints, the NDs loaded with the drug were generally more effective than erythromycin alone in killing streptococci, probably due to ND adsorption and internalization by bacteria (Figure 5).

## 3. Discussion

Wound healing, one of the most multifaceted physiological processes, comprises a wide cohort of cell types tightly controlled over time; thus, the rupture of this step by step involvement could worsen the balance between wound healing and non-healing. Elderly subjects, mainly those affected by diabetes, are at an increasing risk of developing chronic wounds (CWs) that determine severe morbidity and mortality, affecting their quality of life and displaying a health care economic burden for treatments [35,36]. In CWs, a decreased proliferation and an aspect close to that of senescent cells feature the normally present cells (i.e., fibroblasts and keratinocytes). These characteristics typically occur into cells in presence of low concentrations of oxygen, indicating that CWs are hypoxic. Additionally, the presence of bacteria into wounds, mainly *Staphylococcus aureus*, *Pseudomonas aeruginosa*, and GAS, determines a further delay in the healing process of CWs [35].

Much research focused the attention to potentiate the process of CW healing by enhancing the antimicrobial activity of topically applied treatments able to simultaneously reduce infection, promote tissue regeneration, and increase oxygen levels locally [24,32,35,37,38]. The green production of eco-friendly chitosan-based NDs has recently gained popularity, considering that they can be applied as an appropriate tool for pharmaceutical purposes [25,39,40]. The ND exceptional features are affected by physico-chemical properties (i.e., size, shape, and Z potential) and by the loading of both oxygen and antimicrobials that all together conduct to a superior biological performance in biomedical applications.

On these bases, the aim of the present research was the design, the characterization, and the evaluation of the biological properties of different types of low-weight chitosan NDs, both oxygen free and oxygen loaded, added or not with erythromycin as a suitable treatment for *S. pyogenes* infections. The nanoformulation was conceived for a potential clinical application for the local delivery of the drug in infected chronic wounds. 

The designed and prepared OFNDs and OLNDs were characterized by regular and spherical shape, by an average diameter of about 420 nm and they were featured by a positive Z potential, in line with our previous studies [24,25,37]. Of note, the addition of erythromycin into NDs, either OF or OL, did not negatively interfere with their physico-chemical properties. In fact, similar parameters (~425 nm in average diameter, ~0.23 in polydispersity index and ~30.5 mV in Z potential) to those recorded for erythromycin-unloaded NDs were demonstrated. The determination of the Z potential is crucial to establishing the nanoparticle surface charge and thus estimating the physical stability. In published works, the authors revealed that perfluoropentane NDs displayed a diameter ~300 nm and a positive Z potential, as we also did, but no chitosan was included into their ND structure [41,42]. Similarly, Doostan M. and colleagues [43] reported that the obtained chitosan/erythromycin nanoparticles were spherical in shape and with a Z potential of ~15 mV. On the contrary, perfluorohexane chitosan NDs, used in another study [44], were bigger, with respect to the NDs here manufactured and negatively charged. Altogether, the ND physico-chemical parameters available among literature results are difficult to compare due to different components and methods used for their production, and also considering the various fields of their medical applications.

The main step forward achieved in the present study was the erythromycin release profile over time, from Ery-loaded NDs, both OF and OL. First, our data demonstrated that the drug was successfully loaded into the ND core, since no burst effect was observed. Thereafter, a prolonged erythromycin release kinetics was evidenced, from both Ery-OFNDs and Ery-OLNDs, within 26 h. Similar to our work, researchers have looked at the an in vitro release kinetic revealing that erythromycin from loaded nanostructured lipid carriers increased within 24 h of incubation with no initial burst [45]. On the contrary, a sustained release profile, characterized by an initial burst, was saw from clarithromycin loaded chitosan nanoparticles [46]. 

Since a medical topical use of the prepared ND formulations was conceived, the evaluation of their potential cytotoxic effect on eukaryotic cells, specifically keratinocytes, was mandatory. As stated, CWs are characterized by hypoxic condition that determined a delay into the healing processes. For these reasons, the LDH experiments—by assaying all ND formulations—were conducted both in normoxia and in hypoxia, evidencing that the OFNDs and the OLNDs did not display cytotoxic behavior neither in normoxia nor in hypoxia, as we previously revealed [24,25,32]. Moreover, these data well agree with those of Gao Y. et al. (2021), who demonstrated that blank perfluorohexane chitosan NDs were not toxic for human ovarian adenocarcinoma cells [44]. When here the erythromycin was loaded into the NDs (Ery-OFNDs and Ery-OLNDs), they exhibited a favorable biocompatibility, since low percentages of cytotoxicity were obtained (~6%), similar to when also free-erythromycin was tested. Hsiao K.H. et al. (2020) observed that a cell line of keratinocytes (KERTr cells) was not affected in viability by rifampicin and indocyanine green co-loaded perfluorocarbon nanodroplets [47]. Regarding biocompatibility, an interesting paper demonstrated that normal dermal human fibroblastic cells were not impaired in viability when exposed to lipid nanoparticles loaded with essential oils but in a dose-dependent manner [35].

Our microbiological results were firstly accomplished by confocal microscopy to assess if the NDs would interact with *S. pyogenes* or would be internalized by the bacterium to exploit an intracellular negative effect. The images obtained by confocal microscopy demonstrated that after 3 h, and up to 24 h of incubation, both OFNDs and OLNDs seemed uptaken and internalized by *S. pyogenes*. Interestingly, the presence of erythromycin into the NDs did not interfere with their internalization by streptococci. Finally, the direct anti-streptococcal behavior exerted by all of the ND formulations, over time, was obtained by incubating them together and then determining the colony count expressed as CFU/mL. After 24 h of incubation, both OFNDs and OLNDs displayed a strong antibacterial action towards *S. pyogenes*; indeed, the GAS growth was significantly (*p* < 0.0001) reduced from ~10^7^ to ~10^6^ CFU/mL. Literature data and our previous work reported, in fact, that chitosan-containing nanoparticles displayed an intrinsic anti-microbial activity thanks to the well-known properties of chitosan, and a good protracted oxygen release [19,20,21,22,23,24,25]. Moreover, the incorporation of chitosan within the NDs allows a multi-target activity of this natural compound, mainly targeted on the microbial cell walls and plasma membranes, that turns in an increased permeability and in the leak of cytoplasmic contents out of the microbial cell. Additionally, chitosan interacts with DNA impairing mRNA synthesis, and with cytosol proteins blocking their functions [26,27,30,31]. All of these factors play a role in the improved antimicrobial efficacy [15,44].

Since the direct effects of Ery-OFNDs and Ery-OLNDs, respect to free erythromycin, on *S. pyogenes* growth have not been studied so far, we investigated their anti-streptococcal activity by the CFU/mL count. Hence, the most important data pertain the anti-*S.pyogenes* effect of the Ery-OFNDs and Ery-OLNDs respect to free erythromycin, at an increasing time of incubation. At the shortest incubation times, a pronounced effect on the inhibition of the streptococcal growth was registered for the Ery-OFNDs or Ery-OLNDs; we obtained CFU/mL values of 10^4^ in the presence of Ery-NDs vs. 10^5^ CFU/mL of the free erythromycin after 4 h of incubation, and of 10^3^ vs. 10^4^ CFU/mL, respectively, at 6 h. Thereafter at 24 h, a total inhibition of the *S. pyogenes* growth was registered for Ery-OFNDs, Ery-OLNDs, and free erythromycin. These results are consistent with a research demonstrating that rifampicin and indocyanine green co-loaded perfluorocarbon NDs were more active against bacteria respect to free rifampicin [47]. Other literature data regarding the antimicrobial activity of nanocarriers loaded with antimicrobials are available; however, most of them, using the disk diffusion test, are difficult to compare with the data here reported. Despite this methodological issue, several authors demonstrated a wider inhibition halo for the NDs loaded with different antimicrobial agents [39,40,43,45,46], or an action against biofilm [42]. 

Notably, the here produced Ery-OFNDs and Ery-OLNDs were positively charged, as shown by the Zeta potential values, thus they could interact with the negative charged external layers of *S. pyogenes.* In addition, this interaction could be also promoted by other Ery-ND properties (i.e., the small size and the large surface area). Then, confocal microscopy showed that there is a fast interaction and uptake of the Ery-NDs into the bacteria. This early internalization of Ery-NDs, in turn, determined the significant short term killing of streptococci respect to the free-drug conditions; thereafter, at 24 h of incubation, a total killing of GAS was revealed for both Ery-NDs and free-erythromycin. Such a decrease in the *S. pyogenes* proliferation, at 4 and 6 h of incubation, could be crucial in the beginning of the wound infection process when the streptococci are able to quickly proliferate. The targeted delivery of the drug into the bacteria that are prone to grown at the site of infection, is enabled by the antibiotic encapsulation in NDs. The capability of nanoparticulate systems to provide a constant and prolonged antibiotic release locally can increase the antibiotic therapeutic efficacy [48,49]. Therefore, the antimicrobial appropriate doses might be achieved, thereby preventing the resistance mechanism induction and adverse side effects.

## 4. Materials and Methods

### 4.1. Design, Preparation and Characterization of Nanodroplet Formulations

The shell of the nanodroplets (NDs) was made of low molecular weight (LW) chitosan (degree of deacetylation 75–85%, 50–190 KDa, Sigma-Aldrich, Saint Louis, MO, USA), whereas the inner core was of 2H,3H-decafluoropentane (Fluka, Buchs, Switzerland). The NDs were prepared both oxygen-free (OFNDs) and oxygen-loaded (OLNDs), as described in detail in previous work [24,25]. Moreover, erythromycin loaded OF and OL NDs (Ery-OFNDs and Ery-OLNDs) were prepared incorporating erythromycin estolate (Ery at 1.375 µg/mL; Sigma-Aldrich) in the ND decafluoropentane core, using ethanol as a co-solvent to facilitate drug dissolution.

Finally, an UV-C exposure, prolonged for 20 min, was applied to sterilize all the prepared NDs.

All of the manufactured NDs were characterized for physico-chemical parameters, as recently detailed [24,25]. In particular, the ND average diameter, polydispersity index and zeta potential were measured by photocorrelation spectroscopy using a 90Plus Particle Size Analyzer (Brookhaven, New York City, NY, USA) at a scattering angle of 90° and a temperature of 25 °C. Before each measurement, the ND suspensions were diluted in deionized water. The morphology of all the ND formulations was assessed by transmission electron microscopy (TEM) using a Philips CM10 instrument (Eindhoven, The Netherlands). The aqueous suspensions were sprayed on formvar-coated copper grid and air-dried before observation. All of the analyses were performed in triplicate. Additionally, the gravimetric method was used to determine the oxygen content of both OLNDs and Ery-OLNDs, by the quantification of the produced sodium sulfate after the addition of known amounts of sodium sulfite to the samples [24,25]. 

### 4.2. Evaluation of the In Vitro Release Kinetics of Erythromycin Estolate from Erythromycin-Loaded OFNDs and OLNDs

As previously reported in detail [38], a multi-compartment rotating cell—equipped with a donor chamber, containing 1 mL of either Ery-OFNDs or Ery-OLNDs, and receiving compartment divided by cellulose membrane (cut-off = 12,000 Da, Spectra/Por™)—was used to conduct the in vitro erythromycin release kinetic experiments. At different time-points (from 0 up to 26 h) in the receiving phase, phosphate buffer 0.05 M (pH 7.4) was collected and substituted by an equal quantity of fresh phosphate buffer. 

The quantitative determination of the erythromycin amount in the withdrawn samples was performed by high performance liquid chromatography (HPLC) analysis carried out on a Perkin Elmer pump equipped with a spectrophotometer detector (Waltham, MA, USA), set at the wavelength of 215 nm. The release of the drug, during the whole incubation time, was expressed as percentages of release (% of means ± standard deviation, SD) obtained from at least three different experiments [38].

### 4.3. Assessment of the Potential Cytotoxic Activity of Erythromycin Loaded Nanodroplets on Human Cells by Lactate Dehydrogenase Assay 

The evaluation of the potential cytotoxic feature of the erythromycin loaded NDs (Ery-OFNDs and Ery-OLNDs), respect to unloaded ones, was carried out by the lactate dehydrogenase (LDH) assay as reported in a previous research [24]. Briefly, HaCaT cells (Line Service GmbH, Eppelheim, Germany) were seeded in 6-well plates (at ~10^5^ cells/well) in 2 mL/well of supplemented Dulbecco’s Modified Eagle Medium—high glucose (DMEM HG, Sigma-Aldrich) and incubated over-night at 37 °C. Then, human keratinocytes were treated or not (controls) with 10% v/v of OFNDs, OLNDs, Ery-OFNDs, Ery-OLNDs (Ery concentration: 1.375 µg/mL) or free Ery (1.375 µg/mL), and incubated in normoxia (20% O_2_) in a regular CO_2_ cell culture incubator or in hypoxia (1% O_2_) obtained in a hypoxia chamber, for 24 h. Thereafter, HaCaT cells were collected, lysed, and diluted in 0.5 mM of sodium pyruvate and 0.25 mM of the NADH reduced form. The results were examined by determining the absorbance at 340 nm (Synergy HT microplate reader), and cytotoxicity was expressed as the ratio of extracellular to total (intracellular+ extracellular) LDH activity. LDH activities (intracellular and extracellular) were expressed as μmol of oxidized NADH/min/well.

### 4.4. Microbiological Assays 

All of the microbiological experiments were carried out by using a clinical strain of *S. pyogenes* isolated from a human ulcerated wound and cultured on Trypticase Soy Agar (TSA; Oxoid SpA, Milan, Italy). The strain was kindly provided by the Infermi Hospital, Biella, Italy. The strain was stored in a microbank at −80 °C for extended storage, before being used for the microbiological assays [50]. 

#### 4.4.1. Determination of the Uptake of Nanodroplets by *S. pyogenes* by Means of Confocal Laser Microscopy Images

*S. pyogenes* at ~10^9^ Colony Forming Unit (CFU)/mL, was incubated without or with 10% v/v of OFNDs, OLNDs, Ery-OFNDs, Ery-OLNDs—previously labeled with 7% of fluorescein isothiocyanate (FITC, Sigma-Aldrich)—for different incubation times and in agitation at 37 °C [24,32]. After 3 or 24 h of incubation, bacteria were harvested by centrifugation at 3000× *g* at RT for 10 min, washed 2–3× with phosphate-buffered saline (PBS 1x) to remove unbound NDs. After the final wash, each sample was resuspended in 1 mL of PBS 1x and 50 μL of all the NDs were transferred onto separate glass slides, heat-fixed, and then stained for 15 min with 5 μg/mL of a propidium iodide (Invitrogen-Thermo Fisher Scientific Inc., Waltham, MA, USA) solution in a humid chamber at 37 °C. *S. pyogenes* was fixed with mounting solution and covered by cover slips. All ND formulations were observed using an Olympus IX70 inverted laser scanning confocal microscope, and images were recorded by using a FluoView 300 software (Olympus Biosystems, Melville, NY, USA) [24,32].

#### 4.4.2. Evaluation of The In Vitro Anti-*S. pyogenes* Activity of Erythromycin Loaded Nanodroplets 

*S. pyogenes* was cultured at ~10^4^ CFU/mL and subsequently incubated with 10% v/v of UV-C sterilized OFNDs, OLNDs, Ery-OFNDs, Ery-OLNDs or with free erythromycin (1.375 µg/mL), in sterile specimen tubes for 4, 6, and 24 h at 37 °C. The bacteria incubated in Todd Hewitt broth (Oxoid SpA) alone were used as control growth. At each incubation time, from each ND formulation, and from controls as well, a 10-fold dilution was settled up in sterile saline solution and 100 μL were seeded on TSA. All of the agar plates were incubated for 24 h at 37 °C and then the count of CFU/mL was executed [24,25].

### 4.5. Statistical Analysis 

For each type of research, at least three independent experiments were executed, and each condition was performed at least in triplicate. For physico-chemical parameters, results were reported as means ± standard deviations (SD), whereas for LDH and microbiological assays, data were expressed as means ± standard errors of the means (SEM). Confocal images were selected as representative pictures. The one-way analysis of variance (ANOVA) followed by Tukey’s post hoc test, was used for the statistical analysis of the results (GraphPad Software San Diego, CA, USA). *p* < 0.05 was considered significant.

## 5. Conclusions

As observed in our recent studies, low molecular weight chitosan emerged as a good candidate for manufacturing NDs with suitable physico-chemistry, biocompatibility, oxygen release kinetics, and effectiveness parameters [24,25,32,51]. The present study showed the successful loading of erythromycin into the low-weight chitosan-shelled NDs, leading to Ery-NDs that displayed spherical shape, small diameter, cationic surface, and a favorable biocompatibility on human keratinocytes. Moreover, the tested NDs showed greater effectiveness in counteracting streptococcal growth in a long-term manner because of early cellular uptake and internalization. Noteworthy, the most interesting result regards Ery-NDs that were more effective against *S. pyogenes*, respect to free drug, within 24 h of incubation, probably due to Ery-NDs internalization by bacteria and sustained release kinetics of erythromycin into the microorganisms.

Even if we are extremely conscious that converting in vitro observations into in vivo therapeutic regimes is still challenging, based on our findings, oxygen-loaded Ery-NDs appear to be a promising, sustainable, and skin-friendly formulation for the topical management of streptococcal skin infections, favoring the healing of hypoxic wounds, and in the meantime boosting the effectiveness of available antimicrobial drugs.

## Figures and Tables

**Figure 1 ijms-24-01865-f001:**
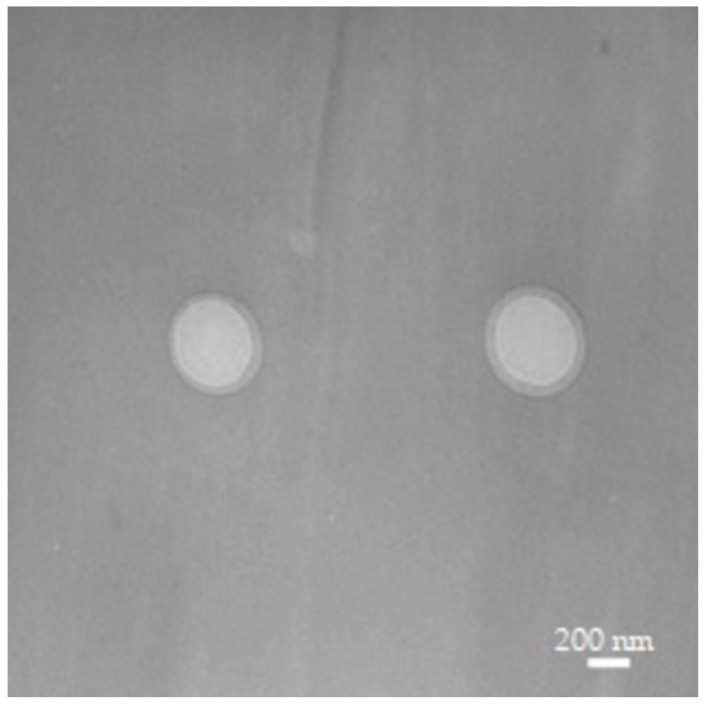
TEM image of low weight chitosan-shelled nanodroplets loaded with erythromycin (Magnification 28,500×).

**Figure 2 ijms-24-01865-f002:**
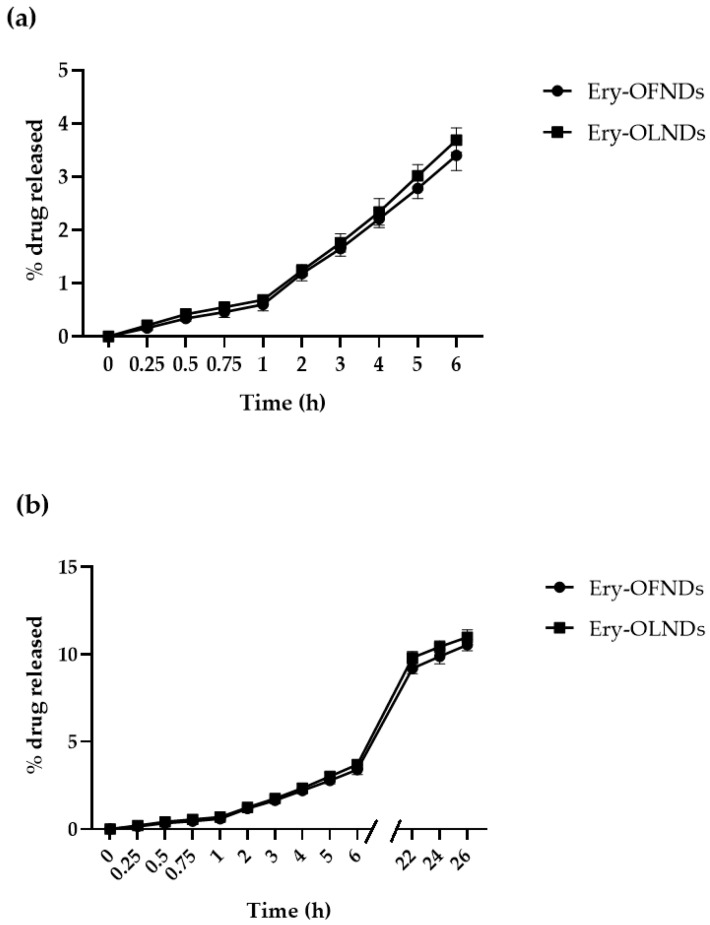
In vitro release kinetics of erythromycin estolate from both the drug-loaded OF and OL nanodroplet formulations: represented as short-term (**a**) and long-term (**b**) percentages of release.

**Figure 3 ijms-24-01865-f003:**
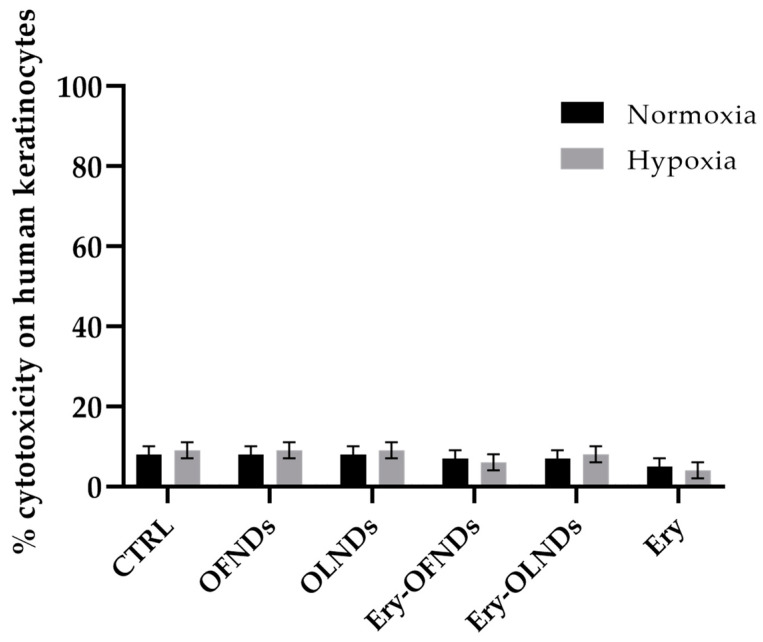
Cytotoxicity of all ND formulations, determined by LDH assay, on human keratinocytes (HaCaT cells), either in normoxia or in hypoxia.

**Figure 4 ijms-24-01865-f004:**
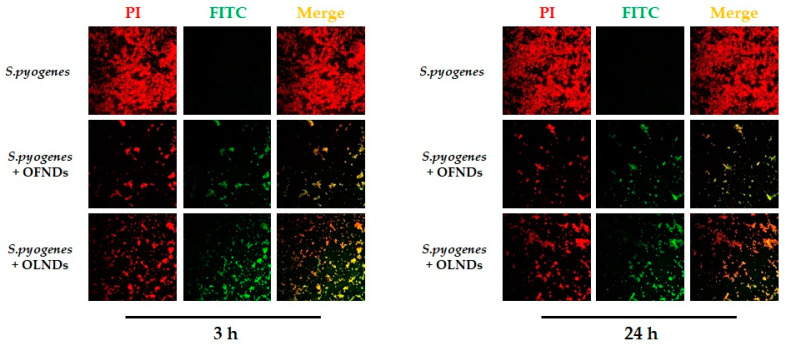
Representative confocal microscopy images of ND internalization by *S. pyogenes* after 3 and 24 h of incubation. Data are shown as representative images from three independent experiments. Red: PI. Green: FITC. Magnification: 100×.

**Figure 5 ijms-24-01865-f005:**
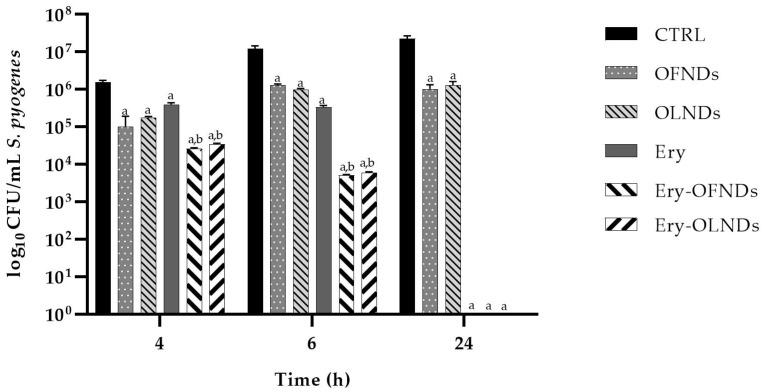
Anti-*S. pyogenes* activity of OFND or OLND formulations, expressed as Log_10_ CFUs/mL, loaded or not with erythromycin. vs. control: *^a^ p* < 0.0001; vs. Ery: *^b^ p* < 0.001.

**Table 1 ijms-24-01865-t001:** Physico-chemical parameters of low weight chitosan-shelled oxygen-free or oxygen loaded nanodroplets added or not with erythromycin.

Nanodroplets	Outer Shell Polysaccharide	Inner Core Fluorocarbon	Fluorocarbon Boiling Point	O_2_ Content (g/mL ± SD)	Average Diameter (nm ± SD)	Polydispersity Index	Zeta Potential (mV ± SD)	Osmolarity (mOsm ± SD)	Viscosity (cP ± SD)
OFNDs	LW-chitosan	DFP	51 °C	/	404.5 ± 22.95	0.22	31.95 ± 3.44	283 ± 0.4	1.33 ± 0.01
OLNDs	LW-chitosan	DFP	51 °C	0.45 ± 0.01	437.3 ± 33.08	0.22	32.07 ± 2.10	282 ± 0.6	1.32 ± 0.02
Ery-OFNDs	LW-chitosan	DFP	51 °C	/	409.8 ± 13.96	0.23	30.15 ± 3.97	285 ± 0.5	1.34 ± 0.01
Ery-OLNDs	LW-chitosan	DFP	51 °C	0.46 ± 0.01	440.2 ± 30.23	0.23	30.82 ± 3.57	285 ± 0.4	1.32 ± 0.02

NDs: nanodroplets; OF: oxygen-free; OL: oxygen-loaded; Ery: erythromycin; LW: low weight; DFP: decafluoropentane; SD: standard deviation.

## Data Availability

The authors confirm that the data supporting the findings of this study are available within the article and/or on request from the corresponding authors (V.A. and M.A.).

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
