# Peer review of "Design, Characterization, and Biological Activities of Erythromycin-Loaded Nanodroplets to Counteract Infected Chronic Wounds Due to Streptococcus pyogenes"

_ijms, 2023, doi:10.3390/ijms24031865_

Round 1
Reviewer 1 Report
The aim of the study was design, characterization and evaluation of the biological properties of different types of low-weight chitosan NDs, both with or free of oxygen, loaded or not with erythromycin. Synthetic NDs were characterized for physico-chemical properties, drug release kinetics, toxicity for human keratinocytes and antibacterial activity against S. pyogenes. The paper is well-written and the topic is very important.
Some notes:
Fig.4: To comply with the rest of the text, LW-OFNDs and LW-OLNDs should probably be replaced with OFNDs and OLNDs, as indicated in line 74 "(LW-cNDs, from now NDs)".
Lines 165-166: Incorrect incubation time - (6, 24 and 48 hours).
The authors claim that "the NDs with the drug were generally more effective than erythromycin alone" (lines 172-173) and "drug-loaded NDs revealed a more potent anti-S.pyogenes activity respect to free erythromycin" (lines 192-193). However, in my opinion, these allegations require corroboration of additional data that are not included in the manuscript.
The activity of NDs with Ery is difficult to compare correctly with the activity of free Ery, as there is no data on the MICs of structures studied, no Ery content is specified in the loaded ND (only v/v NDs concentrations used), no Ery concentrations are specified. It should be noted that after 24h incubation both Ery-NDs and free Ery completely suppressed the growth of microorganisms. Differences in activity on earlier observation periods may be due to different Ery content in the samples. Therefore, it is highly desirable to provide quantitative data on bacteriostatic content.
Line 191: NDs "could perform a targeted delivery of the drug into the site of infection". This statement is not confirmed experimentally, because the infection site and the microorganism are two different things.
Lines 293-294: Assertion of higher doses of bacteriostatic that can be applied with NDs also have no experimental confirmation.
Lines 360-362: It is not clear how the unbound FITC was removed. This is also not described in references 24 and 30.
Reviewer 2 Report
Mandras et. al developed erythromycin-loaded chitosan nanodroplets (Ery-NDs), both oxy-gen-free and -loaded, and studied their efficacy and effectiveness against Streptococcus pyogenes. In addition, evaluated the biocompatibility (in human keratinocytes) and internalization of prepared nanoparticles ( into Streptococcus pyogenes). The study is valuable and noteworthy. I would recommend it for publication after revision.
1. In the introduction, a brief mechanism can be added. To explain the mechanism some examples can be taken from here https://www.ncbi.nlm.nih.gov/pmc/articles/PMC8405884/, https://www.mdpi.com/2079-4991/10/4/643/htm
2. Please correct typo errors on lines 85, 125, and 128 (erythromycin).
3. Please provide more TEM images and SEM can also be used. More characterization studies are good to have.
4. Authors should design and add a mechanism of action of nanoparticles and their action on Streptococcus pyogenes in the discussion part.
5. From lines 168-170 "As detailed in Figure 5, both OLNDs and OFNDs manifested long-term antimicrobial efficacy against S. pyogenes since they significantly (p<0.0001) inhibit bacterial growth up to 24h, thank to the well known chitosan antimicrobial properties." What do authors mean by this? Is it bacteriostatic or bactericidal? Please add appropriate references.
6. With respect to the above comment, what is the perspective of the authors if prepared nanoparticles will only be bacteriostatic? Do authors think it is viable if there is active bacterial growth after 24 hours? Wound healing needs weeks to months if bacteriostatic then it may affect the time of healing as well as how is it feasible?
7. What is the significance of choosing human keratinocytes for biocompatibility studies?
8. Please give numerical values in the abstract and conclusion to evaluate the efficacy and efficiency of nanoparticles against the bacteria used in the study.
9. Add more references in the introduction, results, and discussion. Please check for grammar and typos carefully.
Reviewer 3 Report
This paper reported the design, characterization and biological activities of erythromy-cin-loaded nanodroplets to counteract infected chronic wounds due to Streptococcus pyogenes. NDs and proper Ery-NDs appear to be the most promising and skin-friendly approaches for the topical treatment of streptococcal skin infections allowing wound healing during hypoxia. The manuscript is well-organized and fits the scope of the journal. The references and experiments can support the conclusions in this manuscript. Even though, the revision is required before its publication on Int. J. Mol. Sci.
1. A figure illustration is required for this erythromy-cin-loaded nanodroplets in this manuscript.
2. The previous reported erythromy-loaded nanodroplets are required to be discussed.
3. The concentration/amount of different formulations in the anti-S. pyogenes activity study should be noted.
